# 1 Technical Note: Identifying biomass burning emissions during ASIA-

# 2 AQ using greenhouse gas enhancement ratios

- 3 Jason A. Miech<sup>1,2</sup>, Joshua P. DiGangi<sup>1</sup>, Glenn S. Diskin<sup>1</sup>, Yonghoon Choi<sup>1,3</sup>, Richard H. Moore<sup>1</sup>, Luke
- 4 D. Ziemba<sup>1</sup>, Francesca Gallo<sup>1,3</sup>, Carolyn E. Jordan<sup>1,3</sup>, Michael A. Shook<sup>1</sup>, Elizabeth B. Wiggins<sup>1</sup>, Edward
- 5 L. Winstead<sup>1,3</sup>, Sayantee Roy<sup>1,2</sup>, Young Ro Lee<sup>4</sup>, Katherine Ball<sup>4</sup>, John D. Crounse<sup>4</sup>, Paul Wennberg<sup>4</sup>,
- 6 Felix Piel<sup>5</sup>, Stefan Swift<sup>6</sup>, Wojciech Wojnowski<sup>5,7</sup>, Armin Wisthaler<sup>5,6</sup>
- <sup>1</sup>NASA Langley Research Center, Hampton, 23666, USA
- <sup>2</sup>Oak Ridge Associated Universities, Oak Ridge, 37831-0117, USA
- <sup>3</sup>Analytical Mechanics Associates, Hampton, 23666, USA
- <sup>4</sup>California Institute of Technology, Pasadena, 91125, USA
- <sup>5</sup>Department of Chemistry, University of Oslo, Oslo, 0313, Norway
- <sup>6</sup>Institute for Ion Physics and Applied Physics, University of Innsbruck, Innsbruck, 6020, Austria
- <sup>7</sup>Gdańsk University of Technology, Gdańsk, 80-233, Poland
- Correspondence to: Jason A. Miech (jason.a.miech@nasa.gov)
- **Abstract.** Biomass burning (BB) is a primary source of atmospheric chemistry reactants, aerosols, and greenhouse gases.
- Smoke plumes have air quality impacts local to the fire itself and regionally via long distance transport. Open burning of
- agriculture fields in Southeast Asia leads to frequent seasonal occurrences of regional BB-induced smoke haze and long-range
- transport of BB particles via the northeast monsoon. The Airborne and Satellite Investigation of Asian Air Quality (ASIA-AQ)
- campaign visited several areas including the Philippines, South Korea, Thailand, and Taiwan during a time of agricultural
- burning. This campaign consisted of airborne measurements on the NASA DC-8 aircraft aimed to validate observations from
- South Korea's Geostationary Environment Monitoring Spectrometer (GEMS) and to address local air quality challenges. We
- developed a method that used a combination of BB markers to identify ASIA-AQ DC-8 data influenced by BB and flag them
- for further analysis. Specifically, we used rolling slope enhancement ratios of CO/CO<sub>2</sub> and CH<sub>4</sub>/CO along with mixing ratios
- of CH<sub>3</sub>CN, HCN, and CO, and particle scattering coefficient measurements. The flag was triggered when a combination of
- these variables exceeded a flight specific threshold. We found varying levels of BB-influence in the areas studied, with data
- flagged for BB being <1 % for the Philippines and Korea, and <2 % for Taiwan, but 19 % for Thailand. Our method for
- flagging ASIA-AQ BB-affected data can be used to focus additional analyses of the ASIA-AQ campaign such as pairing with
- back trajectories, satellite hotspot products, and microphysical aerosol characteristics.

#### 1 Introduction

- As Asian economies and populations continue to grow, so will their contribution to global greenhouse gas (GHG) emissions,
- driven primarily by increases in fossil fuel combustion and seasonally by biomass burning (BB) events. Left unchecked, these
- emissions will negatively impact air quality and climate, and therefore it is imperative that emission sources are properly
- identified and accounted for in emission inventories. Sources of methane (CH<sub>4</sub>) can include: fossil fuels (coal mining),

agricultural emissions (enteric fermentation and rice cultivation), and solid waste disposal and wastewater treatment; and for 35 carbon dioxide (CO<sub>2</sub>): fossil fuel/biofuel combustion processes (powerplants and transportation) and industrial processes (cement production) (Kurokawa et al., 2013; Kurokawa and Ohara, 2020). Carbon monoxide (CO), while not a greenhouse 36 37 gas is also emitted from urban sources such as domestic (residential heating and cooking), industrial, and transport sectors 38 (Kurokawa et al., 2013; Kurokawa and Ohara, 2020). Biomass burning is also a significant source of these GHGs and CO; 39 however, the seasonal nature of crop residue burning, and unpredictability of natural fires makes it difficult to accurately 40 quantify these variables sources (Akagi et al., 2011; Streets et al., 2003; Crutzen and Andreae, 1990). The Airborne and Satellite Investigation of Asian Air Quality (ASIA-AQ) field campaign was conducted during February-41 42 March 2024 near the maximum period of seasonal agricultural burning in Southeast Asia, providing an ideal opportunity to 43 further study biomass burning. ASIA-AO was an international joint air quality campaign between the National Aeronautics 44 and Space Administration (NASA) and several space and environmental agencies in Korea (National Institute of 45 Environmental Research (NIER) and Korea Meteorological Administration (KMA)), the Philippines (Department of Environment and Natural Resources (DENR), Philippine Space Agency (PhilSA), and Manila Observatory), Thailand (Geo-46 Informatics and Space Technology Development Agency (GISTDA) and Pollution Control Department (PCD)), and Taiwan 47 48 (Ministry of Environment, National Central University (NCU), and Academia Sinica). Observations included in-situ 49 measurements of trace gases, aerosol properties, radiation fluxes, and meteorological parameters from NASA's DC-8 aircraft 50 and several Korean aircraft, remote sensing measurements from NASA's G-III aircraft, satellite observations from Korea's 51 GEMS instrument, and several ground station measurements. 52 Current observation-based (top-down) techniques for GHG source apportionment include aircraft-based mass balances 53 (Cambaliza et al., 2015), standard eddy covariance (Helfter et al., 2016), positive matrix factorization (Guha et al., 2015), and 54 isotopic signatures (Fiehn et al., 2023). Enhancement ratios between GHGs have also been used to characterize sources of BB 55 plumes (Akagi et al., 2011; Andreae and Merlet, 2001; Nara et al., 2017; Yokelson et al., 1996); however, Yokelson et al. 56 (2013) pointed out several issues with this technique including problems that arise when background concentrations are changing over the measurement period. Halliday et al. (2019) used short-term ΔCO/ΔCO<sub>2</sub> calculated over 60s rolling windows 57 and filtered data by the coefficient of determination (R<sup>2</sup>) to negate the need for a consistent background measurement. DiGangi 58 59 et al. (2021) applied a similar method to data from NASA's Atmospheric Carbon Transport-America (ACT-America) 60 campaign to characterize local CO<sub>2</sub> emissions. DiGangi et al. (2025) adapted this method for use with CH<sub>4</sub>/CO ratios for NASA's Cloud, Aerosol, and Monsoon Processes Philippines Experiment (CAMP<sup>2</sup>EX) airborne field campaign in a chemical 61 62 influence flag, with a focus on separating BB and urban influences. Previous work during the Fire Influence on Regional to 63 Global Environments and Air Quality (FIREX-AQ) airborne campaign used CO and black carbon (BC) enhancements to 64 identify and flag smoke plumes (Warneke et al., 2023). Since the ASIA-AQ campaign overflew many urban areas, a more BB-65 specific approach was required. The method presented in this work combines the use of CO/CO<sub>2</sub> and CH<sub>4</sub>/CO ratios to further 66 constrain source contributions. To specifically target BB sources, we have also incorporated mixing ratios of acetonitrile 67 (CH<sub>3</sub>CN) (Yokelson et al., 2009; Lobert et al., 1990; Holzinger et al., 1999), hydrogen cyanide (HCN) (Yokelson et al., 2009;

- Lobert et al., 1990; Holzinger et al., 1999), CO (Lin et al., 2013; Nara et al., 2017), and aerosol scattering coefficient (Lin et
- al., 2013; Tsay et al., 2013) into our method. This work provides a targeted method to identify air masses influenced by biomass
- burning during the 2024 ASIA-AQ airborne field campaign, an example of which is shown in our Thailand case study.

#### 2 Data and Methods

# 2.1 ASIA-AQ Field Campaign and In Situ Aircraft Measurements

These methods were developed for the ASIA-AQ campaign, which was a joint international field study focused on air quality 74 challenges local to its areas of study, which included the Philippines, Korea, Thailand, and Taiwan. Data for this campaign 75 were collected from aircraft, including NASA's DC-8 and G-III and three Korean aircraft, and several ground sites. Sixteen 76 DC-8 science flights were conducted between 6 February 2024 and 27 March 2024; the full flight break-down can be found in Table S1 along with typical flight paths in Figures S1-4. The NASA DC-8 aircraft carried a variety of instruments used for 77 78 in-situ gas-phase, aerosol, radiation, and meteorological measurements. CO2 was measured via non-dispersive infrared 79 spectroscopy using a modified LICOR 7000 instrument with an uncertainty of 0.25 ppm at <500 ppm and 2 % at > 500 ppm 80 (Vay et al., 2003). CH<sub>4</sub> and CO were measured via wavelength modulation spectroscopy using the Differential Absorption 81 Carbon monoxide Measurement (DACOM) instrument, with CO uncertainties of 2 % for < 1 ppm and 5 % for > 1 ppm and 82 CH<sub>4</sub> uncertainty of 1 % (Sachse et al., 1987, 1991). HCN mixing ratios were measured using a Chemical Ionization High 83 Resolution Time of Flight Mass Spectrometer with an uncertainty of 25 % plus 35 pptv (CIT-CIMS) (Crounse et al., 2006). 84 CH<sub>3</sub>CN measurements were taken using a Proton-Transfer-Reaction Time-of-Flight Mass Spectrometer (PTR-MS) with an 85 uncertainty of 13 % (Müller et al., 2016; Reinecke et al., 2023). Total aerosol scattering at 550 nm measurements were 86 measured using a TSI-3563 Nephelometer with an estimated accuracy of 20 % and estimated precision of 1 Mm<sup>-1</sup> (Anderson 87 and Ogren, 1998). Accumulation-mode aerosol BC mass concentrations were measured by a Single Particle Soot Photometer 88 (SP2-D) (Droplet Measurement Technologies Inc. (DMT), Longmont, Colorado, USA) with an uncertainty of 20 %, operated 89 by NASA Langley Research Center. Optical particle size distributions for particles with diameters between 63.1 and 1000 nm 90 were measured using a DMT Ultra High Sensitivity Aerosol Spectrometer (UHSAS) Model UHSAS-0.055 (DMT, Longmont 91 Colorado USA) with an uncertainty of 20 %, while particles with diameters between 3.16 and 89.1 nm were sized using a 92 Scanning Mobility Particle Sizer (SMPS) with an uncertainty of 20 %. Submicron particle number concentrations (>10 nm) 93 were measured with a TSI Condensation Particle Counter (CPC) CPC-3772 (TSI Inc., Shoreview, Minnesota, USA) with an 94 uncertainty of 10 %. A duplicate instrument also with 10 % uncertainty was used to measure non-volatile particle number 95 concentrations (>10 nm) with the sample passing through a thermal denuder heated to 350 °C prior to entering the instrument. 96 All DC-8 data used in this work had a time resolution of 1 Hz, with the exceptions of the SMPS which measured every minute. 97 Hourly Thai precipitation measurements were obtained from the Thailand Pollution Control Department using the Phaya Thai, 98 Khet Phaya Thai, and Bangkok sites. This data and the developed BB flag are publicly available on the ASIA-AQ data archive 99 (ASIA-AQ Science Team, 2024).

#### 2.2 Rolling Slope Method

The rolling slope calculation method employed here was taken from Halliday et al. (2019), where linear fits over 120 s rolling windows of CO vs. CO<sub>2</sub> and CH<sub>4</sub> vs. CO were calculated using error adjusted bivariate regression as detailed in York et al. (2004) and Cantrell (2008). Applying this method, a maximum of 121 observations were used in each calculation, while the minimum for a valid calculation was set at three. Slopes with low goodness of fit ( $R^2$ <0.5) were filtered out as uncorrelated, and slopes with a  $\Delta$ CH<sub>4</sub>,  $\Delta$ CO, or  $\Delta$ CO<sub>2</sub> less than five times the precision value were dropped. While Halliday et al. (2019) demonstrated that the shapes of the slope distributions are somewhat insensitive to the  $R^2$  cutoff (when  $R^2 \ge 0.5$ ), here we observed differences between correlated slopes ( $R^2 \ge 0.5$ ) and uncorrelated slopes ( $R^2 < 0.5$ ). Halliday et al. (2019) also showed that varying the window width did not drastically change the slope distributions. Similarly, Figures S5-7 show that with increasing window width, the number of correlated slopes increases and with an increasing  $R^2$  cutoff the number decreases; however, the shapes of the slope distributions are similar. Table S2 exhibits the percentage of calculated  $\Delta$ CO/ $\Delta$ CO<sub>2</sub> and  $\Delta$ CH<sub>4</sub>/ $\Delta$ CO slopes that were correlated for each flight. On average across all flights, 52 % of  $\Delta$ CO/ $\Delta$ CO<sub>2</sub> slopes and 53 % of  $\Delta$ CH<sub>4</sub>/ $\Delta$ CO slopes were correlated according to the  $R^2 \ge 0.5$  criterion, in 120 s rolling windows.

 $\Delta$ CO/ $\Delta$ CO<sub>2</sub> rolling slopes can be used to isolate plumes or air mass boundaries with correlated behavior and provide a rudimentary source classification. The combustion efficiency of the source is inversely related to the  $\Delta$ CO/ $\Delta$ CO<sub>2</sub> slope, therefore we have divided up source behaviors into four combustion efficiency bins based on  $\Delta$ CO/ $\Delta$ CO<sub>2</sub> slopes: 0-1 %, 1-2 %, 2-4 %, and >4 %, as previously used by Halliday et al., (2019). In this work, the >4 % combustion efficiency bin is most relevant for identifying BB-influenced plumes. Figure 1a shows an example of this classification applied to the CO and CO<sub>2</sub> bulk ratios from Thailand Flight 1 demonstrating the ability of this method to highlight different air mass behaviors.  $\Delta$ CH<sub>4</sub>/ $\Delta$ CO slopes have been used to distinguish between urban sources of methane and biomass burning (DiGangi et al., 2025; Reid et al., 2023), with higher slopes indicative of more urban sources and smaller slopes 0<x<-40 % with biomass burning. Figure 1b displays the application of these regimes to CO and CH<sub>4</sub> bulk ratios. Both Fig. 1a and 1b demonstrate a large influence from slopes indicative of biomass burning, >4 % for  $\Delta$ CO/ $\Delta$ CO<sub>2</sub> (shown in dark red) and 0<x<-40 % for  $\Delta$ CH<sub>4</sub>/ $\Delta$ CO (shown in orange).

Figure 1. CO vs CO<sub>2</sub> mixing ratios colored by  $\Delta$ CO/ $\Delta$ CO<sub>2</sub> rolling slope enhancement ratio regimes (a) and CH<sub>4</sub> vs CO mixing ratios colored by  $\Delta$ CH<sub>4</sub>/ $\Delta$ CO rolling slope enhancement ratio regimes (b) for Thailand flight 1.

### 2.3 Biomass Burning Flag Determination

The biomass burning flag uses a combination of variables indicative of or a product of biomass burning, including CO mixing ratio, particle scattering at 550 nm, HCN, CH<sub>3</sub>CN,  $\Delta$ CO/ $\Delta$ CO<sub>2</sub>, and  $\Delta$ CH<sub>4</sub>/ $\Delta$ CO. The BB flag is triggered when at least two of the variables exceed their flight-specific thresholds, except for  $\Delta$ CH<sub>4</sub>/ $\Delta$ CO which must fall in a range between zero and its threshold (Table S3). Using a single variable was deemed insufficient due to the possibility of confounding factors, for example, the utility of CH<sub>3</sub>CN as a BB maker has been shown to be less effective in urban areas due to interference from vehicle and solvent usage emissions (Huangfu et al., 2021). However, when CO mixing ratio and particle scattering are paired together a third variable needs to meet its threshold to trigger the flag; the reasoning for this is to prevent false positives occurring from combustion processes other than biomass burning, such as transportation and industrial sources. Additionally, the BB flag is triggered when all four non-slope variables are within 10 % of their threshold to address edge cases. Example scenarios for triggering the flag are shown in Table 1.

Table 1. Example scenarios for triggering the BB flag for Thailand flight 1 (03/16/2024). Underlined values represent the variables that have exceeded their threshold (x), italic values are variables within 10 % of their threshold (x). The last row represents an example where all four non-slope variables are at least within 10 % of their threshold (x).

| BB Flag       | 0<ΔCH <sub>4</sub> /ΔCO< <i>x</i> (%) | ΔCO/ΔCO <sub>2</sub> >x (%) | CO>x (ppm)  | HCN>x (ppt) | CH <sub>3</sub> CN>x (ppb) | Total scattering @ 550 nm>x (Mm <sup>-1</sup> ) |
|---------------|---------------------------------------|-----------------------------|-------------|-------------|----------------------------|-------------------------------------------------|
| Threshold (x) | 44.5                                  | 4                           | 0.32        | 2750        | 1.50                       | 160                                             |
| ✓             | N/A                                   | <u>5.1</u>                  | <u>0.67</u> | <u>2810</u> | 1.31                       | <u>487</u>                                      |
| ✓             | <u>40.3</u>                           | <u>9.8</u>                  | <u>0.34</u> | 1310        | 0.56                       | <u>227</u>                                      |
| ×             | 54.6                                  | 2.5                         | 0.22        | 774         | 0.30                       | 36                                              |
| ✓             | N/A                                   | N/A                         | 0.53        | <u>2823</u> | 0.91                       | <u>386</u>                                      |
| ×             | N/A                                   | N/A                         | 0.33        | 1053        | 0.57                       | <u>186</u>                                      |
| ✓             | N/A                                   | 2.2                         | <u>0.67</u> | 2715        | 1.43                       | <u>418</u>                                      |

#### 2.3.1 Variable Threshold Determination

The BB flag shares similarities with the Chemical Influence flag developed for CAMP<sup>2</sup>EX, where lower positive  $\Delta$ CH<sub>4</sub>/ $\Delta$ CO rolling slopes were associated with biomass burning influence (DiGangi et al., 2025; Reid et al., 2023). Here, the specific  $\Delta$ CH<sub>4</sub>/ $\Delta$ CO cutoff values were chosen on a flight-by-flight basis by evaluating the slope distribution for each flight and looking for distinctive populations. An example of this process is shown in Fig. 2a, where there is a distribution separation occurring at a slope value of 44.5 % for the first Thailand flight. For  $\Delta$ CO/ $\Delta$ CO<sub>2</sub> the threshold was set to greater than 4 % for all flights, as low efficiency processes like biomass burning are associated with higher  $\Delta$ CO/ $\Delta$ CO<sub>2</sub> values (Halliday et al., 2019). The thresholds for the other variables were initially determined by identifying at which concentration within the biomass burning regime (as set by  $\Delta$ CH<sub>4</sub>/ $\Delta$ CO) the variable is distinguishable from non-biomass burning influence regimes. In the Fig. 2b example, that regime is from 0-44.5 %  $\Delta$ CH<sub>4</sub>/ $\Delta$ CO, as determined from Fig. 2a, and the CH<sub>3</sub>CN threshold was set to 1.00 ppb. This method was repeated for the other variables (HCN, particle scattering, and CO mixing ratio) and for all flights. Thresholds for the mixing ratio and scattering variables were set on a flight-by-flight basis. Flight specific thresholds were chosen over campaign thresholds due to changing background concentrations, time periods, and environmental conditions for each flight and location.

Figure 2 (a) Frequency distribution of rolling slope  $\Delta$ CH<sub>4</sub>/ $\Delta$ CO enhancement ratios. The shaded region is the set BB regime for this flight, as determined by the minimum in the distribution. (b) CH<sub>3</sub>CN vs  $\Delta$ CH<sub>4</sub>/ $\Delta$ CO for Thailand flight 1. The blue vertical line represents the edge of the biomass burning influence regime at 44.5 %  $\Delta$ CH<sub>4</sub>/ $\Delta$ CO, while the red horizontal line is the preliminary CH<sub>3</sub>CN threshold for the BB flag. The upper left quadrant of the figure represents the BB regime.

In the case of thresholds that were more difficult to distinguish, a higher value was initially selected to minimize false positives in the flag. Thresholds were then further refined from this initial value by examining the sensitivity of the total percentage of points flagged to that threshold. The final thresholds were chosen at the lowest levels for which the percentage of points remained essentially constant. Figure 3 demonstrates the results from one of these sensitivity tests performed on the first flight in Thailand. In this case, decreasing the thresholds for CO mixing ratio, CH<sub>3</sub>CN, HCN, and particle scattering results in large increases in the number of points flagged. In Fig. 3a, the dashed HCN and CH<sub>3</sub>CN traces are still decreasing past their initial threshold values represented by the triangles (2400 ppt and 1.0 ppb); therefore, this is motivation to increase their thresholds by ~15 % and ~50 % respectively, resulting in the solid traces. The initial thresholds for CO mixing ratio and particle scattering (0.32 ppm and 160), shown as triangles Fig. 3b required no adjustment for this particular flight; the decrease in the percentage of points flagged for the post-adjustment traces is due to the increases in the HCN and CH<sub>3</sub>CN thresholds. The final determined thresholds and rolling slope regimes for all flights are in Table S3, while Table 1 provides some example scenarios of the BB flag being triggered or not for Thailand flight 1. The concentration thresholds varied by a factor of 2.6 for CO and up to 8 for CH<sub>3</sub>CN, this variability can be explained by changing background concentrations for each flight day and location. This approach was followed to minimize false positives in the dataset; therefore, this method is less sensitive to air masses with only minor influences from biomass burning.

Figure 3. Example variable threshold sensitivity plots for the first Thailand flight. (a) shows the sensitivities for the HCN and CH<sub>3</sub>CN thresholds established using plots like Fig. 2b in the dashed traces and the final thresholds in solid. (b) shows the sensitivities for the CO mixing ratio and total particle scattering thresholds established using plots like Fig. 2b in the dashed traces and the final thresholds in solid. The triangles represent the initial thresholds for each variable and the squares the final thresholds.

# 2.4 HYSPLIT Back Trajectories

Air mass history was probed using 48 h back trajectories calculated from the DC-8 flight track at one second intervals using NOAA's HYSPLIT model using GFS 0.25° meteorology (Draxler, 1999; Draxler and Hess, 1997, 1998; Stein et al., 2015). For more specific source type, altitude, and receptor location analysis, we used the BB flag as a filter and focused on specific areas and altitudes of the flight path, such as urban areas at low altitudes to determine whether the emissions measured in these locations were local or transport. To empirically quantify these contributions, we totaled the number of back trajectory points under 1 km that traveled through a certain area prior to ending up at the receptor location; locations used in this analysis are described in Table S4 and shown in Fig. S8.

#### 2.5 VIIRS Fire Hotspots & Imagery

The Visible Infrared Imaging Radiometer Suite (VIIRS) I-Band 375 m Active Fire data product was used to assess fire hotspot density over Southeast Asia (Schroeder et al., 2014). Data version 2.0 from the Suomi National Polar-Orbiting Partnership (Suomi NPP) spacecraft were downloaded from NASA-FIRMS (Fire Information for Resource Management System) (NASA FIRMS, 2024). VIIRS Corrected Reflectance (True Color) imagery was downloaded from NASA-FIRMS (Lin and Wolfe, 2022a, b; NASA VIIRS Characterization Support Team (VCST)/ MODIS Adaptive Processing System (MODAPS), 2022a, b). The Ozone Mapping and Profiling Suite (OMPS) Aerosol Index overlay was downloaded from NASA-FIRMS (Torres and Goddard Earth Sciences Data and Information Services Center (GES DISC), 2019).

#### 3 Results and Discussion

# 3.1 Breakdown of points flagged for each area

Figure 4a breaks down the percentage of points flagged for BB for each flight and for each location, with a breakdown based on cities in Fig. 4b. Very little BB-influence was observed in the Philippines, with most of it occurring north of Manila under 1.5 km above ground level (AGL) (Fig. S1). For Korea, the highest occurrences of BB occurred on flights 2 and 3. On flight 2 the majority of the BB was observed over the West Sea under 500 m AGL. For flight 3 there was BB-influence over both the West Sea and Seoul above 1 km AGL (Fig. S2). Most of the observed BB-influence for the campaign occurred in Thailand, specifically flights 1 and 2, with Chiang Mai experiencing more BB compared to Bangkok (Fig. S3), except for flight 3. Section 3.2 delves more into the geographical and temporal BB flag differences observed in Thailand during the campaign. Taiwan had the next highest percent occurrence, but Korea overall had more points flagged for BB compared to Taiwan. The total time spent collecting data over these locations is the reason for this discrepancy, the campaign spent almost four times as much time over Korea compared to Taiwan. However, in terms of cities, Kaohsiung had the third highest percentage and number of points flagged, after Chiang Mai and Bangkok, with most of these flagged points occurring on flight 1. Most of the BB observed in Taiwan occurred during the third flight, with the majority occurring on parts of the flight track not over Kaohsiung and above 1 km AGL (Fig. S4).

Figure 4. Percentage of points flagged for BB for each flight over each country (a) and select cities (b) with total number of points flagged displayed.

#### 231 3.2 Thailand Case-Study

#### 3.2.1 Flight by Flight Breakdown

Since Thailand had the largest prevalence of biomass burning among the flight locations, we have broken down these flights in greater detail. Figure 5 shows the flight-by-flight breakdown of the BB flag (a-d), VIIRS 48h fire hotspot density maps (e-

235 h) and VIIRS satellite imagery with the OMPS aerosol index overlay (i-l) for the four Thailand flights. While flight 2 had the 236 highest overall occurrence of BB (10,092 points compared to 7,725) (Fig. 5b), flight 1 had more BB-influence closer to the 237 surface (1,289 compared to 1,144) (Fig. 5a). An increase in fire hotspot density is also observed when comparing Fig. 5e to 238 5f, where the density has increased in Eastern Thailand, Cambodia, and Northern Thailand/Myanmar. However, a similar 239 increase is not observed looking at the OMPS aerosol index overlays (Fig. 5i to Fig. 5j). For flight 1 (Fig. 5i) the aerosol index 240 was elevated over northern Laos, northern Vietnam, northern and western Thailand, and central and eastern Myanmar, indicating either a more concentrated, thicker and/or higher layer of absorbing aerosols (dust and smoke) in these areas. For 241 242 flight 2 (Fig. 5j) the higher aerosol index values were concentrated over Hanoi, with western Thailand clearing up and few 243 elevated values over eastern Thailand, Cambodia, and southern Laos. This apparent aerosol index decrease around Thailand 244 contrasts with what was observed using the BB flag and the VIIRS fire hotspot density maps, however, the OMPS aerosol 245 index is also sensitive to changes in aerosol height, concentration, and degree of absorption (Torres and Goddard Earth Sciences Data and Information Services Center (GES DISC), 2019). Prior to flight 3, the area received a considerable amount 246 of rain (~41 mm in Bangkok and up to 82 mm on the northern flight track on the day before flight 3), which likely washed out 247 248 most of the smoke and put out local fires, seen in the decreased fire hotspot density across Thailand (Fig. 5g), and decreased 249 aerosol indices (Fig. 5k). For the final flight, there was a slight increase in the number of points flagged compared to flight 3 250  $(\sim 17 \%)$ , more near the surface as burning resumed, which is apparent when comparing the VIIRS fire hotspot density maps 251 (Fig. 5g and Fig. 5h) and scattered elevated aerosol indices over northern Thailand (Fig. 5l). 252 Figure 5a-d also shows a breakdown of BB-influence in terms of major population centers on the flight track, specifically 253 Bangkok on the southern portion of the track and Chiang Mai on the northern part. For flight 1, no BB-influence was observed 254 below 1 km in Bangkok across four low passes. This contrasts with observations during one pass in Chiang Mai where ~80 % 255 of the low altitude data were flagged for BB. This is consistent with Fig. 5e, where the fire hot spot densities were higher 256 around Chiang Mai compared to Bangkok, and Fig. 5i with elevated aerosol indices over Chiang Mai. For flight 2, there were 257 equivalent amounts of BB-influence over the two cities, but significantly more at the surface in Chiang Mai, however, relative 258 to the time spent at each location over 90 % of high-level data from Chiang Mai were flagged while only ~25 % of high-level 259 Bangkok data were. Even though the fire hotspot density around Bangkok was higher for flight 2 (Fig. 5f) compared to flight 260 1 (Fig. 5e), that BB-influence does not appear to have reached the surface. One possible explanation is that the prevailing 261 southerly winds were blowing that smoke more towards Central Thailand. In the third flight, the rain likely washed out almost 262 all of the smoke around Chiang Mai and quelled many fires as observed in Fig. 5g. However, there was still some BB-influence observed above Bangkok, perhaps by a transported airmass aloft unaffected by the rain. From Fig. 5g, there were still high fire 263

Thailand in the 48 h leading up to flight 4.

hotspot densities occurring in Cambodia and Laos, which could be traced forward to Bangkok using HYSPLIT back

trajectories (Fig. 6a). For the last flight, there was little BB-influence observed in both cities (~1 % total) but still a meaningful

amount across the rest of the flight track (~8 %). Figure 5h shows that fire activities seemed to have resumed throughout

Figure 5. (a-d) Percentage of points flagged for BB for each Thailand flight across the whole flight track, over Chiang Mai, and over Bangkok for above and below 1 km AGL with total number of points flagged displayed, also shown in Table S5. (e-h) 48 h VIIRS fire hotspot density maps over Southeast Asia for March 16, 18, 21, and 25, 2024. The DC-8 flight track is shown in black. The yellow star represents Chiang Mai, and the red star represents Bangkok. (i-l) VIIRS Corrected Reflectance (True Color) imagery with OMPS Aerosol Index overlay over Southeast Asia for March 16, 18, 21, and 25, 2024.

#### 3.2.3 Air Mass Origin Breakdown

To better understand the air mass history at these locations, we used HYSPLIT back trajectory modeling along the flight path to determine where these air masses came from and traveled through (using criteria from Table S4). Back trajectories with an altitude 

Figure 6. HYSPLIT back trajectory (BT) quantification for BB-flagged data over Bangkok for all flights above and below 1 km AGL. The number of BT points under 1 km are shown for each category along with the total number of BT points.

Figure 7. HYSPLIT back trajectory (BT) quantification for BB-flagged data over Chiang Mai for all flights above and below 1 km AGL. The number of BT points under 1 km are shown for each category along with the total number of BT points.

#### 3.2.4 Comparison of Microphysical Aerosol Characteristics

To further illustrate the utility of the BB flag we have used it to compare the integrated size distribution of submicron aerosol particles. Figure 8a shows particle size distributions for BB-flagged data under 1 km in Chiang Mai and unflagged data under 1 km in Bangkok for the same day (flight 2). The BB-flagged data has a unimodal size distribution with a mean particle diameter of 150 nm (accumulation mode) compared to a bimodal size distribution in the unflagged data with a mean particle

diameter 27 nm (nuclei mode). The second and less significant peak in the unflagged accumulation mode (mean diameter 136 nm) data may be the result of background BB presence, demonstrating that this flag is optimized for clear biomass burning influence and may overlook less obvious BB-influence. Several studies have demonstrated that aerosol size distributions in fresh (<1 h) smoke will start at median diameters ranging from 40-150 nm and grow to larger sizes with a decrease in modal width as they age (Hodshire et al., 2021; Janhäll et al., 2010; Reid et al., 1998). Figure 8b demonstrates a comparison between the same location (Chiang Mai < 1 km) but on flights with differing amounts of BB-influence, i.e. flights 2 and 3. While the flight 3 unflagged data still exhibits smaller particle diameters (mean particle diameter 41 nm) than the flight 2 BB-flagged data (mean particle diameter 150 nm), it demonstrates a relatively stronger bimodal distribution compared to flight 2 unflagged data, with a mean particle diameter of 132 nm in the accumulation mode. Even though the presence of this accumulation mode peak and BB markers (CO, HCN, and CH<sub>3</sub>CN) within 20 % of their respective thresholds indicate an influence from BB, the dominant nuclei mode peak and  $\Delta CO/\Delta CO_2$  (0.38 %) and  $\Delta CH_4/\Delta CO$  (104 %) slopes point to urban combustion sources as the dominant influence for this unflagged data. When looking at black carbon mass concentrations, shown in Fig. 9a, the BB-flagged data has consistently higher BC concentrations compared to unflagged data across Thailand for flights 1 and 2. Flights 3 and 4 had the lowest BB-flagged BC concentrations and were higher compared to the unflagged data but were within one standard deviation. This is consistent with Fig. 5c and 5d where flights 3 and 4 had the smallest number of points flagged for BB. The unflagged BC concentrations for flights 1, 2, and 4 are also higher than unflagged flight 3 (but within one standard deviation); therefore, assuming other BC emissions are consistent between the flight days, this is further evidence that the BB flag does not capture all BB-influenced data, specifically BB-influence in the background. Figure 9b shows that the non-volatile number fraction of fine particles for

BB-flagged data is enhanced, with a narrower range compared to unflagged data across all four flights. One explanation may

be that the increased BC mass concentration is elevating the non-volatile number concentration of BB-flagged data in

combination with a decrease in the number concentration of volatile fine particles due to evaporation during transport. Future

work examining the age of the smoke using short-lived BB tracers could provide more information on the concentration of

volatile fine particles in fresh and aged smoke plumes.

Figure 8. Fitted and measured particle size distributions for BB-flagged and unflagged conditions for flight 2 (a) and for Chiang Mai (b) 

Figure 9. Black carbon mass concentrations (a) and non-volatile number fraction CN > 10 nm (b) for BB-flagged and unflagged data for each Thailand flight. Whiskers are representative of one standard deviation, while the boxes represent the interquartile range, the horizontal line the median, and the diamond the mean.

#### 4 Conclusions

Biomass burning and inefficient combustion contribute to poor air quality and greenhouse gas emissions across the globe. Airborne investigations in regions prone to these emissions provide more detailed and focused measurements than typical ground or satellite methods. This work demonstrated a novel approach for identifying biomass burning-impacted airmasses in an airborne dataset. A combination of biomass burning tracers and indicators were used to distinguish biomass burning-impacted airmasses along each given flight track. The Thailand case study demonstrates the efficacy of this flag in determining areas most influenced by biomass burning and in combination with trajectory models, an idea of air mass history. Additionally, a preliminary analysis of physical characteristics of aerosols revealed differences between BB-flagged and unflagged airmasses, including aerosol size distributions, black carbon concentrations, and non-volatile number fraction. These findings,

- while applicable to Asia, demonstrate the value of the method, which can be applied to other field campaigns with similar
- measurements. The utility of the BB flag can be increased in the future with the use of specific volatile organic compounds
- (VOCs) to provide information on the age of the smoke, e.g., primary BB VOCs versus secondary BB VOCs (Liang et al.,
- 2022). Additional development of a boundary layer flag for the ASIA-AQ dataset will improve future work studying the health
- impacts of biomass burning smoke and inefficient combustion at ground level and transport in aloft layers.

# 366 Data Availability

- All data used in this publication are open access and can found in the ASIA-AQ Data Archive
- (https://doi.org/10.5067/SUBORBITAL/ASIA-AQ/DATA001), the FIRMS Archive
- (https://firms.modaps.eosdis.nasa.gov/download/), and the HYSPLIT model is available at
- https://www.ready.noaa.gov/HYSPLIT.php.

#### 371 Author Contribution

- JAM, JPD, GSD, YC, RHM, LDZ, FG, CEJ, MS, EBW, ELW, SR, YRL, KB, JDC, PW, FP, SS, WW, and AW participated
- in the data collection. JAM, JPD, GSD, MS, FG, KB, and WW performed the data analysis and provided feedback. JAM
- prepared the manuscript with contributions from all coauthors.

#### 375 Competing Interests

The contact author has declared that none of the authors have any competing interests.

### 377 Acknowledgments

- The authors would like to thank Jim Crawford and the entire ASIA-AQ leadership team for their support, guidance, and
- feedback. The authors thank NASA's Earth Science Project Office for their logistical support collecting this data. We thank
- Ryan Bennett, David Van Gilst, and Terry Hu for aircraft navigational and meteorological data. The authors thank Dave
- Eckberg and Mauro Rana for their software and hardware support with the DACOM-DLH instrumentation. Jason Miech's
- research was supported by an appointment to the NASA Postdoctoral Program at the NASA Langley Research Center,
- administered by Oak Ridge Associated Universities under contract with NASA. Data collection was funded by the ASIA-AQ
- project under NASA's Tropospheric Composition Program. PTR-ToF-MS measurements aboard the NASA DC-8 during
- ASIA-AQ were partially funded by the Austrian Federal Ministry for Climate Action, Environment, Energy, Mobility,
- Innovation, and Technology (BMK), represented by the Austrian Research Promotion Agency (FFG), through the Austrian
- Space Applications Programme (ASAP 2022, #FO999900547). IONICON Analytik is acknowledged for supplying a FUSION
- PTR-ToF-MS analyzer and providing staff support. The authors gratefully acknowledge the NOAA Air Resources Laboratory
- (ARL) for the provision of the HYSPLIT transport and dispersion model and/or READY website
- (https://www.ready.noaa.gov) used in this publication. We acknowledge the use of data and/or imagery from NASA's Land,

- Atmosphere Near real-time Capability for Earth observations (LANCE) (https://earthdata.nasa.gov/lance), part of NASA's
- Earth Science Data and Information System (ESDIS). We acknowledge the use of data and/or imagery from NASA's Fire
- Information for Resource Management System (FIRMS) (https://www.earthdata.nasa.gov/data/tools/firms), part of NASA's
- Earth Science Data and Information System (ESDIS).

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
