# Peer review of "Technical Note: Identifying biomass burning emissions during ASIA- # 2 AQ using greenhouse gas enhancement ratios"

_EGUsphere, 2025_

## Author Comment (AC1)

Author responses in red. We thank the reviewer for their time and effort in reviewing our manuscript.

This paper utilizes airborne measurements to determine the presence of biomass burning in southeast asia. The detailed analysis of gas- and particulate-phase tracers provides a robust means to identify sources. They then analyze the location and sources of the plumes and the properties of the aerosol emitted. All told, this is a robust analysis and beneficial to the study of biomass burning and pollution in the region. A few major edits are needed however:

**Major Issues**

• Line 133-134: what are these "inefficient combustion processes other than biomass burning" and how would addition of a third variable remove these. For instance, what type of incomplete combustion due you think would produce CO and particles but not HCN or a high CO/CO2. I would think this might be due to mixing of different sources (maybe that are high in CO2).

This statement has been clarified by removing "inefficient" as other combustion processes can lead to elevated CO and particle levels such as heavy trucks and industrial sources. For example, if CO and particle scattering were above the threshold the flag would be triggered even if the  ${\rm CO/CO_2was}$  <4% (indicating an efficient combustion source) and all other variables did not meet their threshold. With the addition of a third variable, this case is avoided as the other markers and ratios are more specific to biomass burning emissions. Line 135-137 now reads "However, when CO mixing ratio and particle scattering are paired together a third variable needs to meet its threshold to trigger the flag; the reasoning for this is to prevent false positives occurring from combustion processes other than biomass burning, such as transportation and industrial sources." Additional references have been included on line 67-68 regarding the use of HCN and  ${\rm CH_3CN}$  as biomass burning markers (Lobert et al., 1990; Holzinger et al., 1999).

• Line 130-139: need more discussion on how this was decided: why only two variables must be over the threshold, why a flight-by-flight determination of a threshold (wouldn't a campaign threshold be better). Was a sensitivity analysis performed?

A sensitivity analysis on the variable threshold determination was performed and is detailed in Section 2.3.1. In regard to the number of variables used we have included the following statement on line 133 "Using a single variable was deemed insufficient due to the possibility of confounding factors, for example, the utility of CH3CN as a BB maker has been shown to be less effective in urban areas due to interference from vehicle and solvent usage emissions (Huangfu et al., 2021)." Since the campaign covered a wide range of areas at different times of the year, we felt it was appropriate to have a flight-by-flight determination of the threshold due to the changing background levels for each flight. We have added the following to line 164 to emphasize this important point "Flight specific thresholds were chosen over campaign thresholds due to changing background concentrations, time periods, and environmental conditions for each flight and location."

 Figure 5: more clearly identify in the figure itself (not the caption) – that the row represents the day. So above A/E/I it should say March 16 (or something like that).

We thank the reviewer for pointing out the unclear labeling in Figure 5. We have added date labels to each row in the figure to clearly indicate the flight day for each figure subset.

Figure 5: If I am reading this correctly: for figure 5A: roughly 27% of the data collected over Thailand was BB, and about 4% of all data points were BB and below 1 km. But, what we don't know from this figure is how much time was spent below 1 km. If only 4% of the time was spent below 1 km, then ALL of the low level data is tagged as BB. But, if 40% of the flight time was spent below 1 km, then 10% of the low level data is tagged as BB. Would separating out each of the columns into two columns make sense?

We thank the reviewer for their suggestion on how to illustrate Figure 5A-D more clearly. We have edited the figure by separating out the altitude levels into columns, displayed the total number of points flagged for each area and altitude, and changed to denominator for each bar to be the total number of points for the specific altitude level in the area of interest as opposed to the total number of points in the area of interest. For example, Fig 5A now reads that ~30% of the high altitude Thailand data was flagged and ~15% of the low altitude Thailand data was flagged. This edited figure now more clearly illustrates that the majority of the data points collected in Chiang Mai at low altitude for the first two flights were flagged for BB. We have also included a table in the SI displaying these point totals, Table S5, and added the following to line 270 "Percentage of points flagged for BB for each

Thailand flight across the whole flight track, over Chiang Mai, and over Bangkok for above and below 1 km AGL with total number of points flagged displayed, also shown in Table S5."

Figure 8a & 8b: why are there no measurements for the BB data below ~12 nm?
Weren't the BB and unflagged using the same instrument s there should be this data.

We thank the reviewer for pointing out this inconsistency in Figure 8. Data was present below ~12nm for the BB trace, however the graph y-axis minimum was mistakenly set to 1 cm³, therefore cutting off the data points which were <1 cm³. We have made this correction to Figure 8.

**Minor Issues & Typos**

• Line 22: I am not sure how apparent "flag them" is maybe clarify by stating "flag them for further analysis"

We thank the reviewer for this suggestion. Line 22 has been edited to "data influenced by BB and flag them for further analysis."

• Line 33: as written it appears you are stating CO is a greenhouse gas (which it is not) because the first two sentences are about GHG emissions. Possibly remove CO from this sentence and put it in its own sentence stating something like "CO (while not a greenhouse gas) is also emitted from urban sources…"

We thank the reviewer for this suggestion. Line 33-38 now reads "Sources of methane (CH4) can include: fossil fuels (coal mining), agricultural emissions (enteric fermentation and rice cultivation), and solid waste disposal and wastewater treatment; and for carbon dioxide (CO2): fossil fuel/biofuel combustion processes (powerplants and transportation) and industrial processes (cement production) (Kurokawa et al., 2013; Kurokawa and Ohara, 2020). Carbon monoxide (CO), while not a greenhouse gas is also emitted from urban sources such as domestic (residential heating and cooking), industrial, and transport sectors (Kurokawa et al., 2013; Kurokawa and Ohara, 2020). Biomass burning is also a significant source of these GHGs and CO;"

Line 59: "(DiGangi et al., 2025) adapted" should be "DiGangi et al. (2025) adapted"
We have made this correction.

• Line 66: "CO mixing ratio" should be "CO" as you already state you are talking about mixing ratios

We have made this correction.

• Line 103-104: "uncorrelated and, slopes" should be "uncorrelated, and slopes" or "uncorrelated and slopes"

We have made this correction. Line 104-105 now reads "filtered out as uncorrelated, and slopes"

Line 137: change "20240316" to an acceptable date format

We have made this correction. Line 150 has been changed to "(03/16/2024)."

• Table 1: should note that the definition of threshold changes. For CH4/CO, values below5 are BB. For everything else, values above the threshold are BB.

We have edited Table 1 and Table S3 to more clearly define the relationship between each variable and its threshold. We edited line 131-133 to more explicitly state how each variable can trigger the flag "The BB flag is triggered when at least two of the variables exceed their flight-specific thresholds, except for  $\Delta CH_4/\Delta CO$  which must fall in a range between zero and its threshold (Table S3)."

Line 281 & 283: I think "and 25 2024." Should be "and 25, 2024."

We have made this correction.

 Figure 2: caption should include which quadrant represents BB – in this case (upper left)?

In the Figure 2 caption we have added the following "The upper left quadrant of the figure represents the biomass burning influence regime."

• Figure 8a: I believe you can put both of the curves on the same scale and still see them so it is not as complicated a figure.

We have made this edit to Figure 8a.

---

## Author Comment (AC2)

Author responses in red. We thank the reviewer for their time and effort in reviewing our manuscript.

Jason Miech and co-authors present a well written manuscript describing an approach at tagging biomass burning air parcels. Their method builds on prior work using short term changes in gas phase species to identify significant biomass burning compositional influence. Both the method refinements and resultant application are important. I recommend acceptance for publication after addressing a minor issue.

In the discussion of Figures 5A and 5B in Section 3.2.1, it would be helpful to add a note indicating the relative number of Thailand low altitude BB-flagged points to both 5A and 5B because it is difficult for the reader to visually discern the difference.

We thank the reviewer for their suggestion on how to better elaborate on the data shown in Figure 5A-D. We have edited the figure by separating out the altitude levels into columns, displayed the total number of points flagged for each area and altitude, and changed to denominator for each bar to be the total number of points for the specific altitude level in the area of interest as opposed to the total number of points in the area of interest. For example, Fig 5A now reads that ~30% of the high altitude Thailand data was flagged and ~15% of the low altitude Thailand data was flagged. This edited figure now more clearly illustrates that the majority of the data points collected in Chiang Mai at low altitude for the first two flights were flagged for BB. On line 232-234 we have included the number of points for each specific case, which are also displayed in Fig 5A-B "While flight 2 had the highest overall occurrence of BB (10,092 points compared to 7,725) (Fig 5B), flight 1 had more BBinfluence closer to the surface (1,289 compared to 1,144) (Fig 5A)." We have also included a table in the SI displaying these point totals, Table S5, and added the following to line 270 "Percentage of points flagged for BB for each Thailand flight across the whole flight track, over Chiang Mai, and over Bangkok for above and below 1 km AGL with total number of points flagged displayed, also shown in Table S5."